# The Phenotypic Spectrum of PRRT2-Associated Paroxysmal Neurologic Disorders in Childhood

**DOI:** 10.3390/biomedicines8110456

**Published:** 2020-10-28

**Authors:** Jan Henje Döring, Afshin Saffari, Thomas Bast, Knut Brockmann, Laura Ehrhardt, Walid Fazeli, Wibke G. Janzarik, Gerhard Kluger, Hiltrud Muhle, Rikke S. Møller, Konrad Platzer, Joana Larupa Santos, Iben Bache, Astrid Bertsche, Michaela Bonfert, Ingo Borggräfe, Philip J. Broser, Alexandre N. Datta, Trine Bjørg Hammer, Hans Hartmann, Anette Hasse-Wittmer, Marco Henneke, Hermann Kühne, Johannes R. Lemke, Oliver Maier, Eva Matzker, Andreas Merkenschlager, Joachim Opp, Steffi Patzer, Kevin Rostasy, Birgit Stark, Adam Strzelczyk, Celina von Stülpnagel, Yvonne Weber, Markus Wolff, Birgit Zirn, Georg Friedrich Hoffmann, Stefan Kölker, Steffen Syrbe

**Affiliations:** 1Division of Paediatric Epileptology, Centre for Paediatric and Adolescent Medicine, University Hospital Heidelberg, 69120 Heidelberg, Germany; jan.doering@med.uni-heidelberg.de; 2Division of Paediatric Neurology and Metabolic Medicince, Centre for Paediatric and Adolescent Medicine, University Hospital, 69120 Heidelberg, Germany; Afshin.Saffari@med.uni-heidelberg.de (A.S.); georg.hoffmann@med.uni-heidelberg.de (G.F.H.); Stefan.Koelker@med.uni-heidelberg.de (S.K.); 3Epilepsy Center Kork, Medical Faculty of the University of Freiburg, 77694 Kehl, Germany; tbast@epilepsiezentrum.de; 4Medical Faculty, University Hospital Freiburg, 79085 Freiburg, Germany; 5Interdisciplinary Pediatric Center for Children with Developmental Disabilities and Severe Chronic Disorders, Children’s Hospital, University Medical Center, 37075 Göttingen, Germany; kbrock@med.uni-goettingen.de; 6Department of Pediatrics, University Medicine Mainz, 55131 Mainz, Germany; Laura.Ehrhardt@unimedizin-mainz.de; 7Pediatric Neurology, Department of Pediatrics, Faculty of Medicine and University Hospital Cologne, University of Cologne, 50937 Cologne, Germany; walid.fazeli@uk-koeln.de; 8Institute for Molecular and Behavioral Neuroscience, Faculty of Medicine and University Hospital Cologne, University of Cologne, 50931 Cologne, Germany; 9Department of Neuropediatrics and Muscle Disorders, Faculty of Medicine, University of Freiburg, 79106 Freiburg, Germany; wibke.janzarik@uniklinik-freiburg.de; 10Clinic for Neuropediatrics and Neurorehabilitation, Epilepsy Center for Children and Adolescents, Schoen Clinic Vogtareuth, 83569 Vogtareuth, Germany; GKluger@schoen-klinik.de; 11Research Institute for Rehabilitation, Transition and Palliation, PMU Salzburg, 5020 Salzburg, Austria; 12Department of Neuropediatrics, University Medical Center Schleswig-Holstein, Christian-Albrechts University, 24105 Kiel, Germany; Hiltrud.Muhle@uksh.de; 13Danish Epilepsy Centre, 4293 Dianalund, Denmark; rimo@filadelfia.dk (R.S.M.); thamm@filadelfia.dk (T.B.H.); 14Institute for Regional Health Research, University of Southern Denmark, 5230 Odense, Denmark; 15Institute of Human Genetics, University of Leipzig Medical Center, 04103 Leipzig, Germany; konrad.platzer@medizin.uni-leipzig.de (K.P.); Lemke@medizin.uni-leipzig.de (J.R.L.); 16Wilhelm Johannsen Centre for Functional Genome Research, Department of Cellular and Molecular Medicine, University of Copenhagen, 2200 Copenhagen, Denmark; joana.santos@sund.ku.dk; 17Department of Biomedical Sciences, University of Copenhagen, 2200 Copenhagen, Denmark; 18Department of Clinical Genetics, Copenhagen University Hospital, Rigshospitalet and Department of Cellular and Molecular Medicine, University of Copenhagen, 2100 Copenhagen, Denmark; iben.bache@regionh.dk; 19Centre for Paediatric Research, University Hospital for Children and Adolescents, 04103 Leipzig, Germany; Astrid.Bertsche@med.uni-rostock.de; 20Neuropaediatrics, University Hospital for Children and Adolescents, 18057 Rostock, Germany; 21Department of Pediatric Neurology, Developmental Medicine and Social Pediatrics, University of Munich, 80337 Munich, Germany; Michaela.Bonfert@med.uni-muenchen.de (M.B.); Ingo.Borggraefe@med.uni-muenchen.de (I.B.); cvstuelpnagel@steinbeis.co (C.v.S.); 22Epilepsy Center, University of Munich, 80337 Munich, Germany; 23Neuropädiatrie, Ostschweizer Kinderspital, 9006 St. Gallen, Switzerland; PhilipJulian.Broser@kispisg.ch; 24University Children’s Hospital Basel, 4031 Basel, Switzerland; alexandre.datta@ukbb.ch; 25Hannover Medical School, Clinic for Pediatric Kidney, Liver and Metabolic Diseases, 30625 Hannover, Germany; hartmann.hans@mh-hannover.de; 26Klinikum Traunstein, 83278 Traunstein, Germany; Anette.Hasse@kliniken-sob.de; 27Department of Pediatrics and Adolescent Medicine, University Medical Center, Georg August University, 37075 Göttingen, Germany; hennekem@med.uni-goettingen.de; 28Children’s Hospital, 84503 Alt-Neuötting, Germany; h.kuehne@kinderzentrum.de; 29Department of Child Neurology, Children’s Hospital, 9006 St. Gallen, Switzerland; oliver.maier@kispisg.ch; 30Neuropädiatrie, Carl-Thiem-Klinikum Cottbus, 03048 Cottbus, Germany; E.Matzker@ctk.de; 31Department of Neuropediatrics, University Hospital of Children, 04103 Leipzig, Germany; Andreas.Merkenschlager@medizin.uni-leipzig.de; 32Children’s Hospital, 46047 Oberhausen, Germany; joachim_opp@eko.de; 33Hospital St. Elizabeth and St. Barbara, 06110 Halle, Germany; s.patzer@krankenhaus-halle-saale.de; 34Department of Pediatric Neurology, Children’s Hospital Datteln, Witten/Herdecke University, 45711 Datteln, Germany; k.rostasy@kinderklinik-datteln.de; 35Kepler Universitätsklinikum, 4020 Linz, Austria; Birgit.Stark@kepleruniklinikum.at; 36Department of Neurology and Epilepsy Center Frankfurt Rhine-Main, Goethe University Frankfurt, 60528 Frankfurt am Main, Germany; strzelczyk@med.uni-frankfurt.de; 37Epilepsy Center, University of Munich, Germany and Paracelsus Medical University Salzburg, 5020 Salzburg, Austria; 38Department of Neurology and Epileptology, Hertie Institute for Clinical Brain Research, University of Tübingen, 72076 Tübingen, Germany; yweber@ukaachen.de; 39Department of Neurology and Epileptology, University of Aachen, 52074 Aachen, Germany; 40Department of Pediatric Neurology, Vivantes Hospital Neukölln, 12351 Berlin, Germany; Markus.Wolff@vivantes.de; 41Genetic Counselling and Diagnostic, Genetikum Stuttgart, 70173 Stuttgart, Germany; zirn@genetikum.de

**Keywords:** PRRT2, BFIS, PKD, PKD/IC, hemiplegic migraine, familial infantile epilepsy, phenotypic spectrum

## Abstract

Pathogenic variants in *PRRT2*, encoding the proline-rich transmembrane protein 2, have been associated with an evolving spectrum of paroxysmal neurologic disorders. Based on a cohort of children with PRRT2-related infantile epilepsy, this study aimed at delineating the broad clinical spectrum of PRRT2-associated phenotypes in these children and their relatives. Only a few recent larger cohort studies are on record and findings from single reports were not confirmed so far. We collected detailed genetic and phenotypic data of 40 previously unreported patients from 36 families. All patients had benign infantile epilepsy and harbored pathogenic variants in *PRRT2* (core cohort). Clinical data of 62 family members were included, comprising a cohort of 102 individuals (extended cohort) with PRRT2-associated neurological disease. Additional phenotypes in the cohort of patients with benign sporadic and familial infantile epilepsy consist of movement disorders with paroxysmal kinesigenic dyskinesia in six patients, infantile-onset movement disorders in 2 of 40 individuals, and episodic ataxia after mild head trauma in one girl with bi-allelic variants in *PRRT2*. The same girl displayed a focal cortical dysplasia upon brain imaging. Familial hemiplegic migraine and migraine with aura were reported in nine families. A single individual developed epilepsy with continuous spikes and waves during sleep. In addition to known variants, we report the novel variant c.843G>T, p.(Trp281Cys) that co-segregated with benign infantile epilepsy and migraine in one family. Our study highlights the variability of clinical presentations of patients harboring pathogenic *PRRT2* variants and expands the associated phenotypic spectrum.

## 1. Introduction

Pathogenic variants in *PRRT2*, encoding the proline-rich transmembrane protein 2 (OMIM*614386), have been identified in different common familial paroxysmal neurologic disorders. First, pathogenic variants in *PRRT2* were found in individuals and families with paroxysmal kinesigenic dyskinesia (PKD, OMIM#128200) [1], soon followed by reports showing that the same variants in *PRRT2* also cause self-limiting sporadic and familial infantile epilepsy (traditional terminology: benign sporadic and familial seizures, BFIS, OMIM#605751) [2], known as Watanabe epilepsy, and the overlapping disorder of paroxysmal kinesigenic dyskinesia with infantile convulsions (PKD/IC, OMIM#602066) [3,4].

While epilepsy in association with *PRRT2* variants usually starts in infancy, movement disorders predominantly occur in adolescence and adulthood. Dystonia is the most common extrapyramidal motor disorder in PKD and PKD/IC, together with paroxysmal chorea and athetosis. Typically, recurrent brief involuntary hyperkinesias of individuals with PKD are triggered by sudden voluntary movements and appear with a higher frequency during stress and anxiety [5,6].

In BFIS, focal and/or bilateral-tonic-clonic seizures start around six months of age in otherwise healthy infants with a self-limited course of disease, usually remitting around the age of two years. Individuals develop normally despite high seizure burden [5,6].

During recent years, other movement disorders and familial hemiplegic migraine (FHM) have been reported in single cases and a small series of patients harboring *PRRT2* variants [5,7,8,9,10,11,12].

*PRRT2* is located on chromosome 16p11.2 consisting of four exons, encoding a protein with 340 amino acids [1]. *PRRT2* is predominantly expressed in presynaptic membranes and the cytoplasm of neurons in the cerebral cortex, basal ganglia, cerebellum, and hippocampus [1,3,13,14]. PRRT2 is a key component of the neurotransmitter release machinery. It is involved in synaptic vesicle exocytosis and Ca^2+^ sensitivity by interacting with proteins of the fusion complex and Ca^2+^ sensors. Functional analysis in knockout excitatory neurons revealed slowing of the kinetics of exocytosis, weakened synaptic transmission, and markedly increased facilitation [15]. Within neuronal networks, loss of PRRT2 leads to an increase of spontaneous and evoked activity and increased excitability of excitatory neurons, establishing the concept of a PRRT2-related synaptopathy [15,16,17,18].

Although more than 70 variants in *PRRT2* have been reported [5], the recurrent variant c.649dupC accounts for nearly 80% of cases. As most variants are nonsense/frameshift or missense variants that are scattered along the entire gene [18], haploinsufficiency of *PRRT2* is the suggested mechanism leading to paroxysmal neurologic phenotypes. There is no clear genotype–phenotype correlation with remarkable intra-familial variability of manifestations [19]. Single reports on bi-allelic loss of PRRT2 have been described in more severe phenotypes of epileptic disorders with intellectual disability, or combinations of paroxysmal disorders [10,20], suggesting a gene dosage effect.

With this study, we aim to highlight the variability of clinical presentations and expand the associated phenotypic spectrum within a cohort of children and their relatives harboring pathogenic variants affecting *PRRT2*.

## 2. Materials and Methods

Patients with infantile epilepsy and a pathogenic variants in *PRRT2* or 16p11.2 microdeletion syndrome (including *PRRT2*) were collected through routine clinical diagnostics from child neurologists and geneticists in Germany and Switzerland within the German Network for Rare Neurological Disorders in Childhood (“Erhebung Seltener Neurologischer Erkrankungen im Kindesalter, ESNEK”) and from collaborating geneticists in Denmark [21]. Inclusion criteria for the core cohort comprised the presence of a likely pathogenic or pathogenic variant in *PRRT2* in relation to infantile seizures. Variant pathogenicity was assessed according to the ACMG (American College of Medical Genetics and Genomics) guidelines [22]. Affected relatives of index patients, in cases of familial occurrence of *PRRT2* variants, were also included (extended cohort). Family members were included, when they presented symptoms that were compatible with PRRT2-related disease and an inheritance pattern compatible with autosomal-dominant transmission. Clinical data were collected using a web-based survey answered by the child neurologists. The survey included detailed questions on neurological disease, treatment, additional symptoms, and complications. Development was assessed in each case by an experienced child neurologist using a three-tiered scale on communication skills (normal, moderately, and severely disabled) and an assessment of the Gross Motor Functions Classification System (GMFCS) and age-appropriate school performance correlating largely with individual intellectual capacities. Descriptive analysis of genetic and phenotypic data was performed with IBM SPSS Statistics version 25. A comparison between groups was analyzed with two-tailed, unpaired Student’s *t*-tests. *p* < 0.05 was considered statistically significant. All patients or their parents gave their written informed consent and the study was approved by the ethics committee of the University of Heidelberg (S-318/2018).

## 3. Results

We collected detailed genetic and phenotypic data of 40 previously unreported patients from 36 families with BFIS and variants in *PRRT2* (core cohort). Data from 62 symptomatic family members of patients from the core cohort were obtained, comprising a cohort of 102 individuals (extended cohort). Of the 40 children in the core cohort, 11 were male (27.5%) and 29 (72.5%) were female. Age at first reported seizures, clinical and genetic diagnosis, seizure freedom and inclusion into the study is summarized in Table 1. There were no significant differences in the age of seizure onset and seizure freedom between male and female individuals. The mean follow-up period was six years.

### 3.1. Molecular Analysis

Mutations in 17 children were found by targeted sanger sequencing of *PRRT2*, while 23 patients were analyzed with an epilepsy gene panel (n = 21) or whole-exome sequencing (WES, n = 2). Suspected microdeletions from analysis with gene panel or WES were additionally confirmed with array-comparative genome hybridization or multiple ligation probe amplification.

The recurrent frameshift variant c.649dupC; p.Arg217Profs* was identified in 30 cases (75%). Other identified variants in *PRRT2* are summarized in Table 2. A microdeletion syndrome 16p11.2 was diagnosed in four patients. One patient was homozygous for the c.649dupC variant. Two further patients harbored a de novo microdeletion 16p11.2 in combination with a second variant in *PRRT2*, leading to compound heterozygosity for PRRT2-related disease (see Table 2).

In the core cohort, 25 cases (62.5%) were of familial origin, six variants (15%) occurred de novo, in nine cases genetic information of parents was not available. Overall, pathogenic variants in *PRRT2* were confirmed in 76 of 102 cases (extended cohort).

In one family with three affected individuals, the novel heterozygous missense variant c.843C>G, p.(Trp281Cys) was identified (see Figure 2A). This sequence change replaces tryptophan with cysteine. The tryptophan residue is located in a highly conserved transmembrane domain of the C-terminal of the protein. This variant is not present in control databases (gnomAD). The variant is predicted as disease-causing (CADD_phred, M-CAP_score, REVEL_score, SIFT, PolyPhen, MutationTaster_score and FATHMM_score). It is classified as a variant of unknown significance (VUS) according to ACMG guidelines (PM2, PM5, and PP3).

### 3.2. Phenotypic Spectrum

Of the 40 patients (core cohort) with BFIS, 27 had isolated infantile-onset epilepsy, nine developed additional movement disorders, and four developed migraines (Figure 1).

### 3.3. Epilepsy

In the core cohort, infantile seizures started at a mean age of 5.7 months (SD 2.7). Bilateral tonic and tonic-clonic seizures were reported in 27 of 40 (67.5%) patients. Focal onset seizures were present in 13 of 40 (32.5%) individuals, with focal motor seizures without and with impaired awareness in four and nine cases, respectively. One individual with a 16p11.2 deletion developed convulsive status epilepticus at the age of nine months. Nine individuals presented with more than one seizure type during the course of the disease. Seizures stopped at a mean age of 14.4 months in most patients (33), while six patients still had ongoing seizure activity at the time of inclusion into the study (mean age 27 months), and one patient developed CSWS. Three individuals, who had been seizure-free showed single seizures later in life (at a mean age of 7.6 years). There was no significant difference in seizure onset age and time of seizure freedom in patients with generalized and focal onset seizures. Seventy percent (28/40) of index patients had a positive family history of infantile seizures with a total number of 47 affected family members with BFIS where identified. The mean number of anti-seizure medications was 1.58 (range 0 to 3).

### 3.4. BFIS Evolving to Acquired Aphasia with Epilepsy (Landau-Kleffner Syndrome)

In a family with five affected individuals with BFIS, one child developed continuous spikes and waves during sleep (CSWS). The variant c.649dupC was found with WES in all affected family members (Figure 2D). Epilepsy manifested at the age of four months with a cluster of seizures characterized by eye deviation and secondary generalization. Later in the course, she presented with focal motor seizures. Seizure frequency in the first years of life was one to three seizures per month. Sleep electroencephalogram (EEG) at the age of 22 months displayed spike and slow-wave complexes in the left centrotemporal region during sleep. At the age of four years and ten months, seizure semiology changed to perioral clonic jerks and ictal speech arrest. Cognitive decline and progressive disturbance of expressive speech and behavior were noted. Cerebral magnetic resonance imaging (cMRI) was unremarkable. A follow-up EEG showed multifocal epileptiform discharges with a spike–wave index (SWI, the pattern of continuous spike and waves during non-REM sleep in relation to a specific period of time [23]) of 80 to 96%. Treatment with levetiracetam and sulthiame was ineffective. The patient responded partially to prednisolone. Starting at the age of eight years seizure frequency decreased and SWI (<30% during sleep EEGs) concomitantly decreased. Mild to moderate cognitive impairment and attention deficit hyperactivity disorder remained as long-term sequelae.

### 3.5. Non-Epileptic Cyanotic Breath Holding Spells

A girl with the maternally inherited c.649dupC variant, who had infantile seizures from six to eight months of age, developed recurrent non-epileptic cyanotic breath-holding spells at the age of 2.5 years. Two episodes with cyanotic breath-holding spells occurred after minor injuries. The child showed a brief period of crying, followed by apnea and cyanosis. She became unconscious and atonic for about 20 to 30 s followed by rapid reorientation without other neurologic symptoms. Her mother had a positive history of cyanotic breath-holding spells until five years of age and of hemiplegic migraine from adolescence.

### 3.6. Movement Disorders

In the core cohort, 9 of 40 individuals (23%) developed movement disorders in addition to infantile seizures. In six of them, PKD started at the beginning of the second decade of life (10 to 14 years). In two individuals, dystonia started already in infancy (see next paragraph). One girl with bi-allelic variants in *PRRT2* presented with episodic ataxia (EA) and non-epileptic myoclonus after minor head trauma (see: episodic ataxia with myoclonus after minor head trauma). In the extended cohort, four additional individuals reported movement disorders (13/102, 13%).

### 3.7. Paroxysmal Kinesigenic Dyskinesia with Infantile Onset

In a single patient of the core cohort, in whom BFIS started at four months of age, dystonia soon followed at the age of five months (Figure 2C). The girl displayed up to 20 episodes per day of short dystonic inclination of the head to the right side, grimacing, elevation of the right shoulder, and supination of the hand/wrists. While seizures were not controlled with sulthiame, freedom of dystonic movements and seizures was seen after a switch to oxcarbazepine. The father of the patient reported similar episodes of dystonia and choreoathetosis that had started in infancy in combination with infantile seizures at four months of age. Dystonia in this case consisted of tilting of the head, grimacing, and limb dystonia triggered by stress or lack of sleep. He was diagnosed together with his daughter and effectively treated with oxcarbazepine. Both harbored the variant c.649dupC in *PRRT2*.

### 3.8. Benign Myoclonus of Infancy

One girl with infantile epilepsy starting at four months of age who was resistant to levetiracetam and valproate additionally displayed daily clusters of head nodding with a maximum frequency of up to five clusters per day from 10 months of age. After identification of the pathogenic variant c.649dupC in *PRRT2*, both seizures and benign myoclonus were controlled by oxcarbazepine.

### 3.9. Episodic Ataxia with Myoclonus after Minor Head Trauma

One girl with BFIS and bi-allelic variations of *PRRT2* (paternally inherited variant c.649dupC and a *de novo* deletion of 16p11.2) displayed three episodes of encephalopathy after mild head trauma at 20 months, at three years, and at four years of age. Encephalopathy, which consisted of mild sleepiness, ataxia with the inability to walk, myoclonus, aphasia and vomiting, was starting 8 to 20 h after minor head trauma. Computed tomography at 20 months was normal. Brain MRI at three years of age during the second episode displayed bi-hemispheric diffusion restriction of the cerebellum 24 h after the head trauma (Figure 3B). Ataxia and encephalopathy resolved within several days without residues, and a control MRI after three weeks showed normal perfusion of the cerebellum (Figure 3C). A similar MRI pattern with an additional T2-hyperintense cerebellum after seven days was seen during the third episode. The same girl was diagnosed with a focal cortical dysplasia in the right parietal lobe (Figure 3A).

### 3.10. Migraine

Four (4/40, 10%) patients of the core cohort developed forms of migraine with additional neurologic symptoms. Two individuals reported hemiplegic migraine consisting of sensory and motor symptoms with dysesthesia, paralysis of extremities, and speech disturbances, while one individual presented episodes of paroxysmal vertigo beginning at the age of 17 months in the context of severe familial hemiplegic migraine (see Figure 2B) and one individual presented with migraine with confusion, change of behavior (anger), dysarthria, dizziness, and visual aura. In the extended cohort, 23 of 102 (23%) individuals from nine families reported migraine, which was described as hemiplegic migraine in 15 of 102 (15%) and as migraine with aura in 8 of 102 individuals (8%).

### 3.11. Development

Prior to onset of seizures, all children developed normally. Despite clusters of seizures and irrespective of the type of mutation, development was described as normal in 38 of 40 individuals (95%) with normal gross motor function, normal cognition, and communication skills. One patient with a deletion affecting exons two to four displayed mild delay of gross motor functions at the age of three years (Gross Motor Functions Classification System (GMFCS) level one: can walk indoors and outdoors and climb stairs without using hands for support, can perform usual activities such as running and jumping, has decreased speed, balance, and coordination) and one patient with a c.649dupC variant had mild language delay at the time of inclusion into the study at three years of age. One patient developed an epileptic encephalopathy with CSWS leading to mild–moderate intellectual disability. Development was reported as normal in all individuals of the extended cohort.

### 3.12. Associated Morbidity and Mortality

None of the index patients was reported with severe complications (e.g., SUDEP), adverse drug events from medication, or associated medical conditions other than the above mentioned. No deaths related to seizures were reported in the extended cohort. In the core cohort, none was reported with medical conditions affecting cardiac, pulmonary, ophthalmologic, orthopedic, rheumatic, dermatologic, and endocrinologic systems. One (1/40) male individual with 16p11.2 deletion syndrome had vesicoureteral reflux (VUR) in early life.

### 3.13. Brain Imaging

Cranial MRI was found to be normal in the majority of affected individuals of the core cohort (38/40, 95%). Periventricular T2-weighted white matter hyperintensities, reminiscent of periventricular leukomalacia were seen in a single patient. One girl with bi-allelic variation (c.649dupC and 16p11.2del) displayed a parietal focal cortical dysplasia and self-limited restricted cerebellar diffusion changes after mild head trauma (see Figure 3).

## 4. Discussion

We present a series of 40 patients from Germany, Switzerland, and Denmark with PRRT2-related benign infantile epilepsy and rarely reported associated comorbidities, further expanding the phenotypic spectrum.

The mutational spectrum in our cohort is in line with previous studies on BFIS and PKD, where the c.649dupC variant was the most prevalent (75% of cases in our series). Similar to previous reports, most *PRRT2* variants described here are of familial origin [5]. In addition to other known variants, a novel missense variant c.843C>G, p.(Trp281Cys) was identified. This variant is predicted as pathogenic (Appendix A) and co-segregated with BFIS and FHM in one family, compatible with PRRT2-related disease (Figure 2A). Another missense variant affecting the same residue (c.841T>C, p.Trp281Arg) has been reported in the literature associated with PKD [24]. The novel variant is classified as a variant of uncertain significance by the guidelines for the interpretation of sequence variants of the American College of Medical Genetics and Genomics. As there are no functional studies of this variant and no additional affected families described, a rating as “likely pathogenic” is not yet possible. Nonetheless, due to the in silico prediction and the previously described pathogenic variant at the same residue, it is considered as probably disease-causing by the authors. As in most other variants, loss of function with haploinsufficiency of *PRRT2* is the suggested mechanism leading to BFIS and FHM in this variant.

Due to the study design, seizures, classified as benign infantile epilepsy occurred in all 40 children. In line with previous studies, a mean age at seizure onset of 5.7 months was found. Seizures were self-limiting in all but one of patients with 16p11.2 microdeletion syndrome, who developed a convulsive status epilepticus at the age of nine months. Overall, status epilepticus seems a rare feature of PRRT2-associated epilepsy. It was only reported in a single case with a point mutation in *PRRT2*. None were reported in a large cohort of patients with 16p11.2 microdeletion syndrome. [5,12,25,26].

One child with a familial c.649dupC variant developed CSWS at the age of four years. So far, CSWS has not been described in association with *PRRT2*. The incidence of CSWS is estimated 0.2 to 0.5% of all childhood epilepsies [27]. CSWS has been described in various genetic conditions, including common epilepsy-related ion channel genes like *SCN2A* or *KCNQ2*, as well as the N-methyl D-aspartate receptor subunit gene *GRIN2A*, that show overlapping expression at neuronal synapses [28,29]. As no other genetic alteration on WES was identified, it seems possible that the *PRRT2* variant contributed to the development of CSWS in this child.

Irrespective of seizure burden and independent from the pathogenic gene variation, the overall development was described as normal in 95% of patients and affected relatives. Behavioral issues were not reported, which is in line with previous reports [30]. Behavioral comorbidities were not assessed systematically. Interestingly, development, seizures, and additional symptoms did not differ in individuals with heterozygous and bi-allelic variants of *PRRT2*. Further reports are needed, as bi-allelic loss of *PRRT2* has been described previously to result in more severe epileptic disorders with intellectual disability and movement disorders [10,20]. In this study, none of the index patients was reported with severe complications (e.g., SUDEP), adverse drug events from medication, or relevant associated medical conditions. So far an increased risk for respiratory disturbance or SUDEP cannot be associated with PRRT2-related epilepsy despite a single report on SUDEP in one patient [31], which would warrant further studies. Most patients became seizure-free during the second year of life, which is in line with other studies [5]. Nonetheless, several patients were not seizure-free at the time of inclusion into the study or had occasional seizures later in life, suggesting long-term seizure susceptibility in single individuals with PRRT2-related BFIS.

Brain imaging was normal in the majority of cases. Of note is one case of a girl with a compound heterozygous variant that displayed a focal cortical dysplasia, a feature that has not been reported before. A causative relation remains unclear and future studies may help to confirm this finding.

One child was reported with cyanotic breath-holding spells after infantile seizures had ceased and treatment had been tapered. Interestingly, in this family, the variant c.649dupC co-segregated with cyanotic breath-holding spells in the mother. It remains to be elucidated whether pathogenic *PRRT2* variants contribute to susceptibility to breath-holding spells, since one-third of children with breath-holding spells have a positive family history and their genetic origin has not yet been unraveled [32]. Considering that breath-holding spells affect 0.1 to 4.6% of otherwise healthy young children [33], the co-occurrence might be incidental. Interestingly, another case of breath-holding spells in 16p11.2 microdeletion syndrome has been reported, however, in both cases, no complications or life-threatening respiratory symptoms occurred [34].

Being present in 10% of index patients with BFIS, migraine was a prevalent additional paroxysmal neurological phenotype in our study. The frequency was significantly higher than in the general population [35,36], especially considering the relatively young mean age of 6.6 years. In a representative German pediatric cohort a prevalence of 2.4% in the age group 0 to 17 years and 0.1% in the age group 0 to 6 years was reported [37]. This difference further confirms an association of migraine with *PRRT2* and underlines an elevated risk for migraine in patients presenting with BFIS in association with PRRT2. Of note, single cases in our cohort responded well to treatment with sodium-channel blockers, suggesting that the identification of affected patients and family members might be helpful for each individual.

In addition to one previous report of a child with a *PRRT2* variant and benign myoclonus of early infancy presenting as episodic head drops [12], we report an additional case of this rare infantile disorder. Another child in our study displayed early-onset dystonia of limbs and face. Both cases show that PRRT2-associated non-epileptic paroxysmal movement phenomena can occur as early as in infancy and a thorough workup is needed in these young children to distinguish epileptic and non-epileptic disorders.

One girl with a bi-allelic variant in *PRRT2* presented with episodic ataxia. In contrast to other forms of episodic ataxia associated with *KCNA1* (EA1) or *CACNA1A* (EA2), ataxia occurred after minor head trauma and acute cerebellar diffusion restriction together with late T2-hyperintensities was visible on MRI (Figure 3B). Additional and distinct symptoms in this girl during episodic ataxia were temporary myoclonus of hands and face and dysarthria. To date, one adult case with acute-onset ataxia and transient cerebellar diffusion restriction associated with a heterozygous *PRRT2* mutation was published [38]. Additional evidence for a role of *PRRT2* in episodic ataxia comes from previous reports of patients with bi-allelic variants that were associated with a more severe presentation that included combinations of neurological disorders, such as BFIS, PKD, episodic ataxia, developmental delay, and cerebellar atrophy [10,20]. Fortunately, in contrast to previously reported cases, development in our girl remains age-appropriate at four years, despite the bi-allelic loss of *PRRT2* and three episodes of ataxia from mild head trauma. In line with other forms of episodic ataxia, an overlap with headache disorders and epilepsy is present in PRRT2-associated neurological phenotypes and a common synaptic dysfunction might underlie these synaptopathies [16,17].

## 5. Limitations

Our study is based on a cohort of children with primary epilepsy phenotypes in relation to *PRRT2* variants. This selection might reduce the variability of PRRT2-associated neurological phenotypes in this cohort. Thus, estimates in this group do not represent the overall incidence of clinical presentations but may give an estimate of the likelihood of additional symptoms in patients presenting with BFIS. As we included symptomatic family members with known or putative mutations in *PRRT2* in our extended cohort, estimates of phenotypic variability in this group might be more representative of PRRT-related neurological disorders. Due to the retrospective character of our study and the limited availability of genetic data from all relatives we are unable to provide additional analyses on intrafamilial variability and penetrance. Regarding specific manifestations such as migraines, we tried to overcome this fact by concentrating on our core cohort and compared the incidence with reported data. Considering that the disease penetrance is reduced in PRRT2-associated disorders, a cohort of symptomatic individuals may overestimate common phenotypes and still underestimate mild and rare phenotypes.

## 6. Conclusions

Our study on a representative group of patients presenting with BFIS highlights the variability of phenotypic presentations associated with variants of PRRT2 and provides additional insight into the spectrum of PRRT2-related neurological disorders. We present additional evidence for a role of *PRRT2* in early-onset movement disorders, as well as childhood-onset episodic ataxia. In addition to known variants, we report the variant, c.843G>T, p.(Trp281Cys) that co-segregated with benign infantile epilepsy and migraine in one family. With our study, we provide important additional findings from a representative large cohort with detailed clinical data.

## Figures and Tables

**Figure 1 biomedicines-08-00456-f001:**
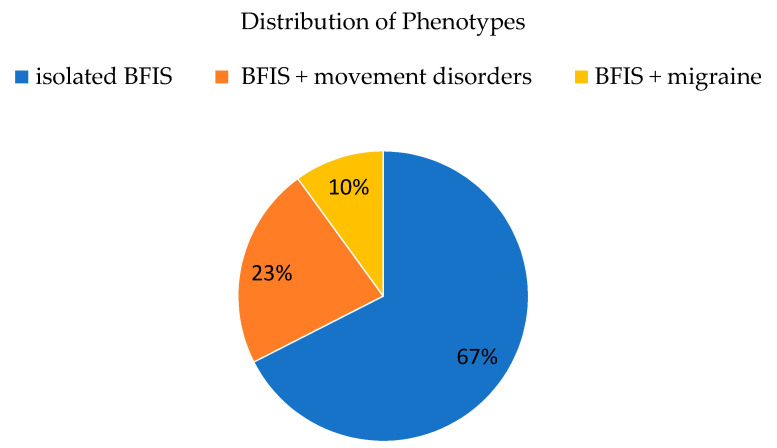
Distribution of clinical manifestations in a cohort with BFIS (core cohort, n = 40).

**Figure 2 biomedicines-08-00456-f002:**
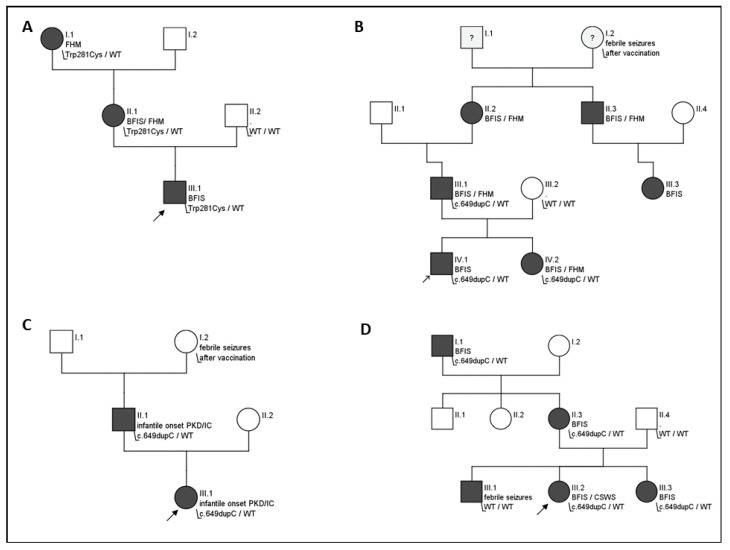
Pedigrees of affected families. Squares: male individuals, circles: female individuals, black circles/squares: clinically affected individuals, blank circles/squares: healthy individuals. Arrows indicating index patient. WT: wild-type; (**A**) family with the novel variant c.843C>G, p.(Trp281Cys) in three members displaying BFIS and/or FHM. Development of the index patient prior to seizure onset had been normal. Epilepsy gene panel diagnostic revealed a heterozygous missense variant at c.843C>G, p.(Trp281Cys). The variant was confirmed in the maternal grandmother. The mother had a history of infantile seizures and developed hemiplegic migraines later in life. The grandmother presented with hemiplegic migraine; seizures were not reported. No other movement disorders or neurologic disease were reported in the family; (**B**) family with five individuals with BFIS and severe hemiplegic migraines that started exceptionally early in two individuals. The index patient presented episodes of paroxysmal vertigo beginning at the age of 17 months in addition to benign infantile seizures consisting of clusters of bilateral tonic seizures beginning at four months of age. Under therapy with oxcarbazepine, the patient’s symptoms disappeared almost completely. The father and three relatives in the father’s family share a history of BFIS and hemiplegic migraine. The father has hemiplegic migraine with speech disturbance. The sister of the index patient developed BFIS without migraine. The index patient, his father and sister harbor the familial c.649dupC pathogenic variant; (**C**) family of two individuals presenting infantile PKD/IC. Seizures and dystonia started in the first half-year of life. The index patient and her father harbor the familial c.649dupC variant; (**D**) family of five affected individuals with BFIS and febrile seizures co-segregating with the familial c.649dupC pathogenic variant, one child developed continuous spikes and waves during slow sleep (CSWS).

**Figure 3 biomedicines-08-00456-f003:**
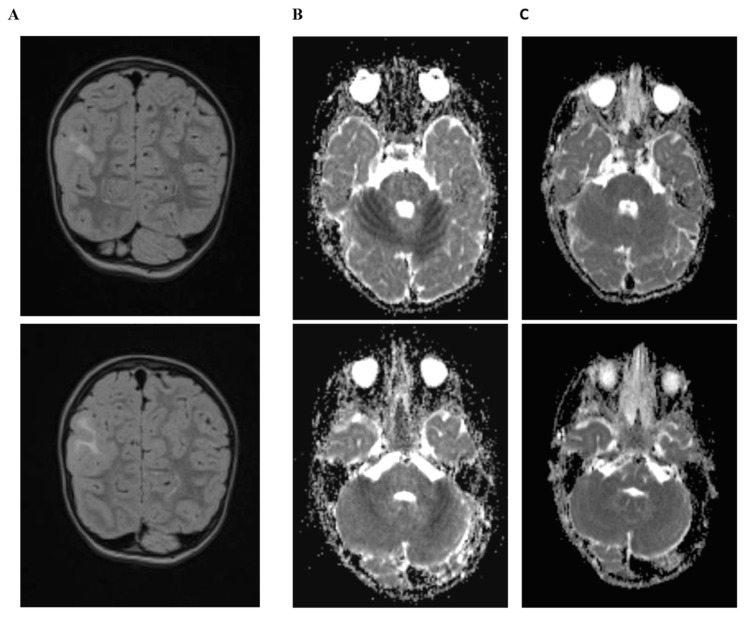
Brain imaging of a girl with bi-allelic variation in *PRRT2* (c.649dupC and de novo deletion of 16p11.2) displaying focal cortical dysplasia in the right parietal lobe with preserved upper cortex layers (**A**) on T2-fluid attenuated inversion recovery MRI. After mild head trauma the girl presented with ataxia, problems of expressive language, and myoclonia lasting one to three days. Diffusion-weighted (ADC, apparent diffusion coherent) imaging revealed hypointense areas in both cerebellar hemispheres (**B**) that normalized after three weeks (**C**).

**Table 1 biomedicines-08-00456-t001:** Characteristics of 41 children with infantile epilepsy and genetic variation of *PRRT2* (core cohort).

	n	Mean	Minimum	Maximum	SD
Age at inclusion (years)	40	6.6	1	26	5.46
Age at first reported seizure (months)	40	5.7	1	15	2.67
Age at seizure freedom (months)	33	14.4	1	132	22.24
Age at diagnosis (months)	40	6.1	1	15	2.85
Age at genetic diagnosis (months)	40	45.3	3	216	60.85

SD: standard deviation.

**Table 2 biomedicines-08-00456-t002:** Genetic variants of 40 index cases (core cohort) and familial cases from the extended cohort.

	Index (n)	Family (n)	Gene (Coding DNA)	Protein
**Heterozygous Variants in *PRRT2***				
	1	0	c.323_324del	p.(Thr108Serfs*25)
	1	1	c.341_342del	p.(Val114Glufs*19)
	1	1	c.593_594del	p.(Pro198Argfs*26)
	30	31	c.649dupC	p.(Arg217Profs*8)
	0	1 ^#^	c.836C>T	p.(Pro279Leu)
	1	1	c.843G>T	p.(Trp281Cys)
	1	0	c.(?-65)_(1243-?)del	exon 2-4 del
	2	0	16p11.2del	16p11.2 del
**Biallelic Variants in *PRRT2***				
Compound heterozygous	1	0	c.649dupC 16p11.2del	p.(Arg217Profs*8) 16p11.2 del
Compound heterozygous	1	0	c.836C>T 16p11.2del	p.(Pro279Leu) 16p11.2 del
Homozygous	1	0	c.649dupC c.649dupC	p.(Arg217Profs*8) p.(Arg217Profs*8)

^#^ Compound heterozygous index patient, missense variant inherited from affected father.

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
