# Peer review of "The Phenotypic Spectrum of PRRT2-Associated Paroxysmal Neurologic Disorders in Childhood"

_biomedicines, 2020, doi:10.3390/biomedicines8110456_

Round 1

Reviewer 1 Report

The authors have been generally responsive to my critiques and comments. I have no further issues.

Reviewer 2 Report

Thank you for your revisions.

I feel you adequately responded to my queries.

I understand this population generally has normal development and that further neuropsychological testing was not performed in this cohort. However, it would have been interesting to see if there were any subtle deficits or certain profiles that could be detedted. Maybe for a future study. 

This manuscript is a resubmission of an earlier submission. The following is a list of the peer review reports and author responses from that submission.

Round 1

Reviewer 1 Report

This manuscript by Doring et al describes the phenotypic spectrum of a new cohort of patients with epilepsy due to PRRT2 mutations. PRRT2 is a known epilepsy gene and many variants have previously been described. In this work, the authors propose to expand the phenotypic spectrum associated with PRRT2 mutations by studying 41 childhood patients and 63 of their extended family. Overall, the paper is well written and it provide useful clinical genetic information by adding to the existing knowledge of the relationship between PRRT2 and epilepsy. However, the rationale and significance of the study could be strengthened to make it more clear how the findings of this study significantly advance clinical knowledge in the field. My major and minor concerns are listed below.

Major concerns:

  1. The purpose/desired outcome of the study needs to be made more clear. The last sentence of the Introduction says the aim was "describing rare clinical phenotypes." However, other places it sounds like the aim was to expand the phenotypic spectrum or highlight the clinical variability associated with PRRT2. From the Introduction, it sounds like most of the genotypes and phenotypes identified in the study are already known so what is the novelty that this study provides? How does this study differ from those previous studies? What is this adding?
  2. For the novel variant that was identified, it would be nice to have some description or discussion of the nature of the mutation (ie, what part of the protein is affected, is this predicted to be loss of function etc, is this a conserved residue). Similar discussion should be added for the common c.649dupC variant.
  3. Discussion should be added to speculate/explain why there is so much phenotypic variability even in patients that have the exact same mutation, such as the c.649dupC variant.

Minor concerns:

  1. in the abstract, 3rd sentence from the end, was "slow-wave" sleep meant instead of "slow" sleep?
  2. in the Introduction, line 87, it should read "knockout excitatory neurons" rather than "knock-down"
  3. in the last sentence of the Methods, a reference should be added for the ACMG guidelines.
  4. in the methods, more details need to be provided for the t-test such as how many tails and paired or unpaired.
  5. in section 3.1, a reference or info should be added regarding the epilepsy gene panel to indicate the genes included
  6. in Table 2, the spacing and placement of the "heterozygous" and "balletic" variant headings should be made consistent and probably move to line up with the left margin
  7. was there a difference in seizure onset age/freedom for the generalized and focal seizure types described in section 3.3?
  8. Fig 3 is cited before Fig 2 in the text. 
  9. the spacing of the lettering in Figure 2 needs to be fixed
  10. in the Fig 3 legend, description of the circles and squares needs to be provided as to whether they indicate male or female
  11. it seems that the long breath holding observed in one patient could suggest risk for SUDEP as identified previously in one PRRT2 study by Labate et al Epilepsy Res 2013, 104:280-4. discussion could be added along these lines.
  12. based on this and other studies, are most PRRT2 disease mutations inherited or de novo?
  13. throughout, there are places where commas are incorrectly placed such as after the first words of sentences like "Although,..." or "Considering,..." 

Reviewer 2 Report

This is a thorough overview describing a cohort of patients with childhood epilepsy and affected family members with PRRT2. It is of great interest to pediatric neurologists managing these patients and delineates and expands the phenotype.

  1. How were developmental assessments performed on patients? Were neuropsychological evaluations performed? Was it obtained restrospectively from qualitative descriptions.
  2. I would be curious to know if any of the patients had other neuro-psychiatric co-morbidities, such as depression, anxiety, ADHD, behavioral issues, increase need of special assistance at school, etc.
  3. What was the mean duration of follow-up of patients and range?
  4. It would be interesting to known the mean number of anti-seizure medications required to obtain control of seizures. How easy was it to control the movement disorder with medication?
  5. Any thoughts about the pathophysiology of the FCD in the one patient with a bi-allelic variant.Did her seizures appear to have a semiology consistent with the FCD? or incidental? EEG show frequent discharges in that area?

Reviewer 3 Report

Döring and co-workers described the phenotypic spectrum of PRRT2-associated

paroxysmal neurologic disorders in childhood. 

Overall, the paper lacks  originality as PRRT2 in childhood has been extensively studied, and phenotype has been deeply described. However some cases described by authors are of interest, specially compound heterozygous cases. 

The main criticism is that including only symptomatic cases and symptomatic family members is quite difficult to calculate the rate of each phenotype in a low-penetrance condition as PRRT2. Authors partially agree in the Limitations section, but they should have made some extra efforts.

For example, it could be interesting calculate intra-family variability and penetrance for each phenotype to add some novelty to the study.

I'm not convinced that all phenotypes described can be related to PRRT2 and need more specification: migraine (without aura) is too frequent to be surely associated to PRRT2, otherwise migraine with motor aura may be. The same consideration for Benign myoclonus or episodes of apnoea during crying.

Finally, I found that variant c.115C>A p.Ala39Thr is not classified as damaging  (I checked only PolyPhen 2); did the authors checked all mutations? in the table 2 should be specified for all variants.
